# Anxiety in Attachment and Sexual Relationships in Adolescence: A Moderated Mediation Model

**DOI:** 10.3390/ijerph19074181

**Published:** 2022-03-31

**Authors:** Alessandra Santona, Alberto Milesi, Giacomo Tognasso, Laura Gorla, Laura Parolin

**Affiliations:** Department of Psychology, University of Milano-Bicocca, 20126 Milan, Italy; alessandra.santona@unimib.it (A.S.); a.milesi6@campus.unimib.it (A.M.); g.tognasso@campus.unimib.it (G.T.); l.gorla8@campus.unimib.it (L.G.)

**Keywords:** attachment anxiety, sexual anxiety, anxiety, internalizing problems, adolescence

## Abstract

Adolescence is characterized by several transformations, such as identity construction, progressive estrangement from parents, relational interest in peers, and body changes that also involve sexuality issues. In this process, attachment patterns play a fundamental role in relationships, and when these are dysfunctional, they can result in internalizing and externalizing problems. Often, females show their relational difficulties through internalizing expressions and males through externalizing expressions. Additionally, given the sexual progress involved in this life moment, psychological symptomatology may influence adolescents’ perception of sex and performance. Our purpose is to study the mediating role of internalizing and externalizing symptomatology in the relationship between attachment patterns and sexual and psychological dimensions. In addition, we investigated the moderating effect of the sex assigned at birth on this mediation model. *n* = 493 adolescents (38.3% males; Mage = 16.51; SD = 1.17) participated in the study. The results show a significant mediation effect of internalizing symptomatology on the relationship between attachment and sexual anxiety. Additionally, this effect is moderated significantly by assigned-at-birth sex. These results confirm that in adolescence, attachment patterns can influence adolescents’ perception of sex. The connection between these two psychological dimensions is influenced by symptomatologic expression. Further investigations are needed.

## 1. Introduction

Adolescence represents the life cycle stage during which individuals build their identity by completing specific developmental tasks [1]. One of these tasks is gradually shifting the attachment bond from parents to peers. Adolescents, typically, are separating from their parents as primary attachment figures, and relying more upon the support of peers [2]. Adolescents begin to use peers and romantic partners as sources of support and intimacy [3]. Early romantic experiences and relationships are an essential component of this stage of life. These experiences can affect the development of the adolescent’s identity and their general sense of competence, or inflict feelings of humiliation that could weaken their self-esteem. From these experiences, adolescents can gain what Bowlby called a safe haven and a secure base, or they could feel that safety and security are precarious and unreachable [2,4,5].

Most of the literature on romantic relationships agrees that two principal dimensions are involved. Brennan, Clark and Shaver [6] defined these dimensions as attachment-related anxiety and avoidance. The anxiety dimension involves fear of abandonment and rejection by romantic partners; the avoidance dimension involves how much a person feels uncomfortable depending on—mentally and physically—and being close to others [7].

### 1.1. Attachment and Anxiety Symptoms

Adolescence is a challenging moment in which emotional dysregulation can be a risk factor for the development of internalizing problems, such as depression or anxiety [8]. This kind of dysregulation is often associated with relational issues which, in this life phase, are already precarious for the changes involving attachment systems [8].

As mentioned above, adolescents are becoming distanced from their parents and are investing more in peers [2]. Attachment has also been identified as a possible protective or risk factor, depending on the quality of the attachments, for the development of psychological symptoms [9]. In this way, it is possible to highlight how attachment relationships could influence the occurrence of anxiety symptoms in adolescence [10,11]. Although some studies have evaluated the relationship between some attachment dimensions and internalizing problems [11,12], no evidence in the literature seems to support the idea of a relationship between attachment-related anxiety and these psychopathological outcomes.

However, the possibility of being rejected by partners or peers, at a moment in which the adolescent is invested mostly in these relations, could result in anxious symptomatology. Moreover, different authors have shown how internalizing symptoms, such as anxiety, could be influenced by the sex assigned at birth [13].

### 1.2. Anxiety and Sexuality

Several studies conducted in the past few years have underscored that the relationship between anxiety and sexual functioning is complex [14]. There are still many unanswered questions about the relationship between sexual satisfaction and anxiety [15,16].

Women with anxiety disorders reported worse sexual functioning than those without anxiety disorders [17] and greater sexual inhibition, because of the threat of performance failure and its consequences. Dispositional anxiety and related worries significantly predicted various types of sexual dysfunctions.

### 1.3. Attachment and Sexuality

A small part of the literature in this field suggests that sexual difficulties are common among adolescents. In one study of Canadian adolescents aged 16–21 years, 50% of participants experienced sexual problems; of those, 50% reported a significant level of distress [18]. In this study, low desire and orgasm difficulty were the most reported sexual problems among women, while low desire and erectile dysfunction were the most reported sexual problems among men.

An increasing amount of evidence has highlighted that attachment orientation could impact sexual experiences. Differences concerning how people experience romantic and sexual relationships could be affected by individual differences in both anxious and avoidance dimensions of attachment [19].

People with a secure attachment style (low on anxiety and avoidance) seem to have long, stable, and satisfying relationships characterized by high investment, trust, and friendship [20]. In contrast, since they develop a negative image of themselves and an attitude of mistrust toward others, insecure relationships with people could be characterized by frustration and dissatisfaction [21,22].

Hazan, Zeifman, and Middleton [23] found that a secure attachment was related to the enjoyment of various sexual activities, including mutual initiation of sexual activity and entertainment of physical contact.

More precisely, people with anxious attachment seem to depend on others and have conflicting relationships. Their relationships are characterized by an obsessive desire for intimacy, fear of abandonment, and not being loved [24]. Attachment anxiety seems to be related to distress about sexual attractiveness, and is also linked to the affectionate and intimate aspects of sexuality, rather than to genital elements. In adolescents [25,26,27,28], anxious attachment is linked with more presexual activity, and less sexual satisfaction and communication [29,30,31]. Specifically, people with an anxious attachment tend to have earlier and more frequent sexual experiences, especially girls [2]. Individuals with an anxious attachment also have a more negative self-view regarding their sexual attractiveness, and more negative sexual cognitions in general [29].

Adolescents with avoidant attachment are mainly detached from emotions in sexuality [25,30]. Avoidant attachment is linked with the avoidance of intimacy, the avoidance of any presexual and sexual activity, or multiple superficial and short-term relationships [29,30,31,32,33]. Individuals with an avoidant attachment seem emotionally distant, independent, and aloof [34,35,36].

The sexual behavior of people with high levels of insecure attachment is often associated with sexual functioning and satisfaction [37]. Avoidant and anxious attachment was consistently related with poor sexual satisfaction, sexual dysfunction, and sexual anxiety. In women, both anxious and avoidant attachment affected sexual dysfunction, but avoidant attachment is an essential predictor demonstrated across multiple studies [38,39,40,41,42]. The same patterns have been evidenced in men. Recent research indicated that anxious attachment, but not avoidant attachment, is significantly related to sexual anxiety and sexual functioning in men [40,43].

Regarding sexual satisfaction, the results are varied, with some studies highlighting links between sexual dissatisfaction and insecure attachment in both men and women [25,44], and others finding no significant associations between sexual satisfaction and attachment insecurity in men [45]. In general, it seems that both attachment avoidance and anxiety influence sexual functioning and sexual anxiety in both men and women. The conflicting results reported in the literature indicate that attachment insecurity is significantly related to sexual anxiety, but the dimensions of avoidance and anxiety may differently impact sexual relationships.

### 1.4. Attachment and Sexual Anxiety

Janda and O’Grady [46] described sexual anxiety as “a generalized expectancy for nonspecific external punishment for the violation of, or the anticipation of violating, perceived normative standards of acceptable sexual behavior”. In men, sexual anxiety is positively related to anxious but not avoidant attachment, and is negatively related to erectile functioning, desire, and sexual dissatisfaction [47]. In women, sexual anxiety is positively related to both anxious and avoidant attachment and sexual dysfunction [47]. These correlations indicate that anxiety could be a mediator between attachment and sexual functioning. These findings are also confirmed by Brassard and colleagues [38] and Davis and colleagues [30]: in both studies, sexual anxiety mediated the relationship between avoidant and anxious attachment and emotional aspects of sexual satisfaction for males and females.

Although many studies have highlighted the possible association between attachment insecurity, in both anxious and avoidant dimensions, and sexual functioning, numerous elements of this association are not well understood. It is possible that attachment insecurity could be correlated with intrapersonal variables associated with sexual difficulties. Modern conceptual models of human sexual behavior reveal possible cognitive factors that may mediate the relationship between attachment and sexual functioning.

### 1.5. Aims and Hypothesis

For this reason, we decided to explore the possible relationship between the anxiety dimensions of attachment, sexual anxiety, and anxious symptomatology.

In Hypothesis 1, we hypothesize that demographic variables, attachment anxiety, anxious symptomatology and sexual anxiety are associated. In Hypothesis 2, we hypothesize that anxious symptomatology plays a mediating role in the relationship between attachment anxiety and sexual anxiety (see Figure 1).

In Hypothesis 3, we hypothesize that anxious symptomatology plays a mediating role in the relationship between attachment anxiety and sexual anxiety, and this mediation effect is moderated by the sex assigned at birth (see Figure 2).

## 2. Materials and Methods

### 2.1. Participants

The participants included 493 students (Mage = 16.85, SD = 1.41) from all five grades from three high schools in Legnano, Italy. We assessed only for sex assigned at birth and not for gender identity, resulting in a sample of 61.7% females (*n* = 304).

Table 1 shows other sample characteristics, such as parents’ marital status (PMStatus), the presence of romantic relationships in students’ lives (RomRel), the presence of longstanding significant relationships (SigRel), and the presence of a significant friend (SigFr).

### 2.2. Measures

Participants completed a battery of self-report tests made up of three instruments: the Attachment Style Questionnaire [48,49], the Brief Symptoms Inventory [50], and the Multidimensional Sexuality Questionnaire [51].

Attachment Style Questionnaire: The ASQ is a 40-item self-report scale developed to assess several dimensions linked to attachment style, such as confidence, discomfort with closeness, need for approval, preoccupation for relationships, and relationships, as secondary. Moreover, this measure assesses avoidance and anxiety in attachment relationships. Participants are asked to rate aspects about them or others on a 6-point Likert scale (item example “I often wonder whether people like me”). The ASQ showed adequate reliability and construct validity in secondary student samples [48,49].

Here, we focus on the attachment anxiety subscale, in which a low score is presumed to represent a higher level of attachment anxiety in relationships.

Brief Symptoms Inventory: The BSI is a 53-item self-report scale developed to assess psychiatric disorder symptoms. Participants are asked to rate whether and how much they were distressed by different symptoms over the last week on a 5-point Likert scale (item example “during the past seven days, how much were you distressed by trouble concentrating”). BSI showed adequate reliability and construct validity [50].

For the study’s purposes, we focused on the anxiety subscale, in which a higher score is presumed to represent a greater level of anxious symptomatology.

Multidimensional Sexuality Questionnaire: The MSQ is a 59-item self-report scale developed to assess several psychological dimensions linked to individual sexual life. Participants are asked to rate how much the item represents their sexual characteristics on a 5-point Likert scale (item example: “I feel anxious when I think about the sexual aspects of my life”). Out of the 12 dimensions obtainable by the MSQ, we focus on the anxiety scale, in which a higher score is presumed to represent a stronger and wider feeling of anxiety about sexual relationships. The MSQ showed adequate reliability and construct validity [51].

### 2.3. Procedure

The study was approved by the Research Ethics Committee of the University of Milano-Bicocca. After obtaining informed consent from the schools and parents, well-trained data collectors disseminated information about the survey to all the participants. The same data collectors went to the abovementioned schools and administered the questionnaires in paper sheet format to students who participated voluntarily.

### 2.4. Statistical Analyses

All statistical analyses were conducted by employing SPSS software (version 27.0.1) and also using the PROCESS macro plug-in for mediation and moderation analyses [52]. Missing values of demographic data are shown in Table 1; no missing data were found in the self-report measures.

First, descriptive statistics and correlations among variables were analyzed.

The hypothesized mediation model (see Figure 1) was tested in a single model using a bootstrapping approach to assess the significance of the indirect effects of the mediator [52]. In the model, we tested the mediation effect of anxious symptomatology on the association between attachment anxiety and sexual anxiety. The “PROCESS” macro, Model 4, v3.5.3 [52] with bias-corrected 95% confidence intervals (*n* = 10,000) was used to test the significance of the indirect (i.e., mediated) effects.

The hypothesized moderated mediation model (see Figure 1) was tested in a single model using a bootstrapping approach to assess the significance of the indirect effects at differing levels of the moderator [52]. In the model, we tested the moderating effect of sex on the association between attachment anxiety and anxious symptomatology in the previous mediation model. The “PROCESS” macro, Model 7, v3.5.3 [52] with bias-corrected 95% confidence intervals (*n* = 10,000) was used to test the significance of the indirect (i.e., mediated) effects moderated by sex, i.e., conditional indirect effects. This model explicitly tests the moderating effect on the predictor-to-mediator path (i.e., path a). An index of moderated mediation was used to test the significance of the moderated mediation, i.e., the differences in the indirect effects across levels of sex [53]. Significant effects are indicated by the absence of zero within the confidence intervals.

## 3. Results

### 3.1. Preliminary Analyses

Means, standard deviations, and zero-order correlations for each variable included in the study are shown in Table 2 and Table 3. Significant positive associations were found between attachment anxiety and anxious symptomatology, as expected from our hypothesis.

Additionally, as expected, anxiety symptoms and anxiety about sex were positively associated. In the end, attachment anxiety and anxiety about sex were positively associated.

### 3.2. Testing for Mediation

The hypothesized mediation model was tested using the PROCESS macro model number 4, which tests a model whereby anxious symptomatology mediates the effect of attachment anxiety on sexual anxiety (Figure 3) [52].

The model was significant, showing an association between attachment anxiety and anxious symptomatology (β = 0.14, *t* = 6.76, CI = 0.10, 0.18). Moreover, attachment anxiety was positively associated with anxiety about sex (β = 0.40, *t* = 3.42, CI = 0.17, 0.63). Additionally, anxious symptomatology was positively associated with sexual anxiety (β = 0.74, *t* = 3.1, CI = 0.27, 0.1.2). Overall, the total effect of attachment anxiety on anxiety about sex was significant (β = 0.50, *t* = 4.48, *p* < 0.001).

### 3.3. Testing for Moderated Mediation

The hypothesized moderated mediation model was tested using the PROCESS macro model number 7, which tests a model whereby sex moderates the effect of path a (Figure 4) [52].

Sex was found to moderate the effect of attachment anxiety on anxious symptomatology (β = 0.10; *t* = 2.60; *p* = 0.01). Higher anxious symptomatology was also associated with higher anxiety about sex (β = 0.74; *t* = 3.10; *p* < 0.005).

The overall moderated mediation model was supported with the index of moderated mediation = 0.08 (95% CI = 0.006; 0.20). As zero is not within the CI, this indicates a significant moderating effect of sex on attachment anxiety, with an indirect effect via anxious symptomatology [53]. The conditional indirect effect was stronger in females (β = 0.14; 95% CI = 0.03; 0.26) and weakest in males (β = 0.06; 95% CI = 0.007; 0.14). Tests of simple slopes (i.e., conditional effects on path a found a weaker association between attachment anxiety and anxious symptomatology for males (β = 0.08; *t* = 2.79; *p* < 0.05) than for females (β = 0.19; *t* = 7; *p* < 0.001)).

## 4. Discussion

Attachment quality as a protective or risk factor for developing mental disorders has been studied widely in recent decades. However, as psychopathological outcomes can vary and multiple variables are involved, no previous study has investigated the model we propose here. The present study investigated the relation between attachment anxiety, anxious symptomatology, and anxiety about sexual relationships. Specifically, our findings examine a complex theoretical model, which includes the mediating role of anxiety symptomatology and the moderating role of the sex assigned at birth in the association between attachment anxiety and anxiety about sexual relationships.

First, positive independent correlations were found between demographics, attachment anxiety, anxious symptomatology and sexual anxiety, supporting the idea that these dimensions move together.

Second, the significant results of the mediation model (Figure 3) support Hypothesis 2, in which anxious symptomatology mediates part of the total effect that attachment anxiety has on anxiety about sexual relationships.

Third, the significant results of the moderated mediation model (Figure 4) support Hypothesis 3, in which assigned-at-birth sex moderates the effect of attachment anxiety on anxious symptomatology, which also mediates the total effect of attachment anxiety on anxiety about sexual relationships.

Our results are in line with the literature. In fact, according to Stefanou [47], sexual anxiety in boys is positively related to anxious attachment and negatively related to erectile functioning, desire, and sexual dissatisfaction. In girls, sexual anxiety is positively related to anxious attachment and sexual dysfunction [47,54]. Moreover, in girls, anxious attachment may lead to poorer sexual functioning through associations with vulnerability to anxiety related to sexual experiences, and the facilitation of the adoption of negative beliefs and attitudes toward sexuality. In addition, Dettore and colleagues [17] stated that there is a relationship between pathological anxiety and sexual anxiety in both boys and girls. People with an anxiety disorder reported worse sexual functioning and a greater propensity for sexual inhibition, because of the threat of performance failure and its consequences. It seems that internal working models (IWMs) could play a key role in defining the relationship between attachment anxiety, anxious symptoms, and sexual anxiety. People with a higher level of anxious attachment could have a negative image of themselves as unlovable and unattractive, and could develop an attitude of distrust of or reliance on others [55,56]. Moreover, their relationships are characterized by a fear of abandonment and not being loved [24]. Scholars have suggested that these kinds of IWMs could, on the one hand, lead to anxiety symptoms [17] and, on the other hand, be correlated with a higher level of sexual anxiety [25,37]. The results we obtained permit us to hypothesize that attachment anxiety impacts anxious symptomatology, which in turn raises the level of sexual anxiety. Our results seem to indicate that anxious symptoms could be a mediator between attachment and sexual anxiety.

### Limitations and Future Research

This study briefly explores the connection between some dimensions that may be central in adolescence; specifically, attachment and sexual anxiety are investigated for their connection to internalizing problems, such as anxiety symptoms. As adolescence is still lacking literature that focuses on this life moment, this study represents an innovative contribution to research on how attachment, sexual life and psychological symptoms are associated in this challenging phase. Although some studies have investigated these variables independently, no study has discussed the results in a complex model, as proposed in this article.

However, some observations about the possible limitations of the study need to be drawn. The sample considered here is wide enough to support our results, but it is characterized by a larger proportion of female adolescents. A future study could match the number of male participants to the number of female participants to understand if the moderating effect of the sex assigned at birth found here could be influenced. Additionally, attachment-related dimensions are assessed with self-report measures, which are very effective time-saving instruments, but which cannot capture the real complexity of such a variable. Indeed, future studies about the relationship between attachment and other variables could be conducted with the Adult Attachment Interview [57].

## 5. Conclusions

This study provides preliminary relevant results about the relationship among attachment characteristics, symptomatic expression, and sexual life dimensions. Here, we provide evidence of the relevant role of anxiety in attachment relationships, which could presumably be considered a risk factor for anxiety in sexual relationships, and how this is influenced by anxiety symptoms expressed mostly by female adolescents.

Further investigations are needed to evaluate the role of other attachment dimensions in developing psychological symptoms, and how these could interact with sexual life, which is a very important dimension of adolescents’ lives.

However, these findings allow us to partially understand how the attachment changes involved in adolescence are linked to some psychological difficulties, and how these are reflected in other adolescents’ tasks, such as challenges with upcoming sexual life.

## Figures and Tables

**Figure 1 ijerph-19-04181-f001:**
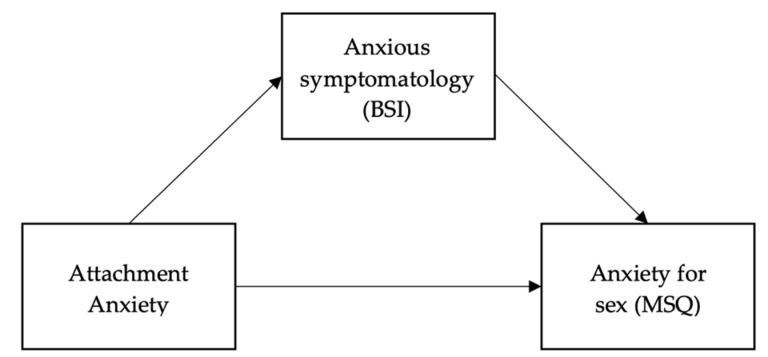
Hypothesis 2 mediation model.

**Figure 2 ijerph-19-04181-f002:**
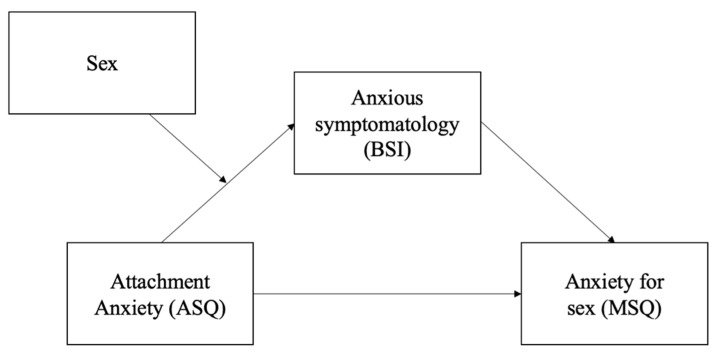
Hypothesis 3 moderated mediation model.

**Figure 3 ijerph-19-04181-f003:**
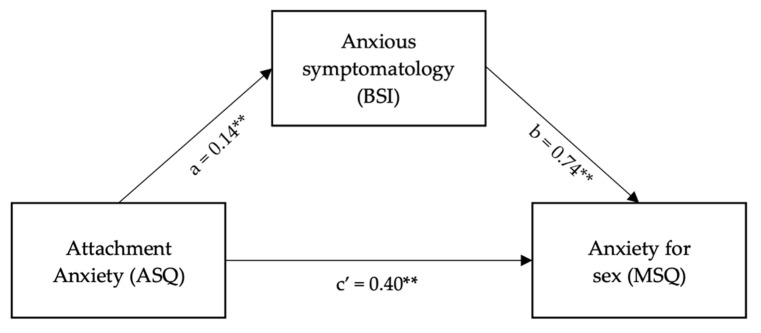
Results of Hypothesis 2 mediation model. ** *p* < 0.001.

**Figure 4 ijerph-19-04181-f004:**
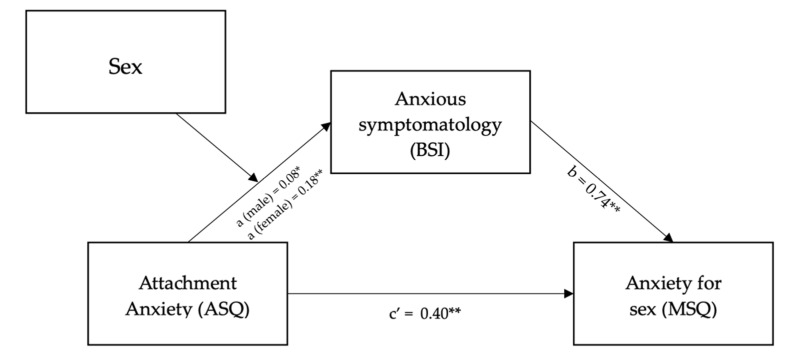
Results of Hypothesis 3’s moderated mediation model. * *p* < 0.05; ** *p* < 0.001.

**Table 1 ijerph-19-04181-t001:** Sample demographics.

	N	%
Gender		
Male	189	38.3%
Female	304	61.7%
PMStatus		
Married	439	89%
Divorced	53	10.8%
Missing	1	0.2%
RomRel		
Yes	185	37.5%
No	307	62.3%
Missing	1	0.2%
SigRel		
Yes	256	51.9%
No	219	44.4%
Missing	1	0.2%
SigFr		
Yes	412	83.6%
No	66	13.4%
Missing	15	3%

**Table 2 ijerph-19-04181-t002:** Mean and SDs of the study variables.

	M	SD
ASQ-Anx	−0.72	1.6
BSI-Anx	1.95	0.78
MSQ-Anx	5.34	4.06

**Table 3 ijerph-19-04181-t003:** Correlations between the main variables of the study.

	ASQ-Anx	BSI-Anx	MSQ-Anx
ASQ-Anx	-		
BSI-Anx	0.29 **	-	
MSQ-Anx	0.20 **	0.19 **	-

Note: ASQ-Anx—attachment anxiety; BSI-Anx—anxious symptomatology; MSQ-Anx—anxiety for sex; ** *p* <0.001.

## Data Availability

The data presented in this study are available on request from the corresponding author.

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
