# Peer review of "Anxiety in Attachment and Sexual Relationships in Adolescence: A Moderated Mediation Model"

_ijerph, 2022, doi:10.3390/ijerph19074181_

Round 1
Reviewer 1 Report
The paper offers insight into the under-examined issue of adolescent sexual relationships by examining attachment anxiety and anxiety for sex and the mediating and moderating roles of anxiety symptomatology and sex assigned at birth. However, the following criticisms are made:
- Clarity, succinctness, and written expression throughout entire paper can be greatly improved on (including typos, comprehension of sentences, etc.)
- Line 143: Authors state "In hypothesis 1, we wanted to test if variables were associated". Please specify what variables you are testing.
- Additionally, hypotheses are actual hypothesis statements but are descriptions of the study's statistical analyses. Please reword the hypotheses as proper hypothesis statements.
- In the hypothesis, statistical analysis, and results sections, authors explicitly state that x "represents the independent variable", y "represents the dependent variable", etc. This form of written expression detracts from the readability of the paper and can be better succinctly written as a single statement (for example, It is hypothesised that anxious symptomatology plays a mediating role in the relationship between attachment anxiety and sexual anxiety)
- Line 150 and throughout the rest of the paper: The authors keep stating that "gender" is a moderator in the relationship model. However, in the methods section, authors state that gender identity was not assessed but sex assigned at birth was actually assessed. Please be more transparent in your representation of the data and variables assessed.
- Table 1: Various demographic data were collected (e.g., age, presence of significant relationships) but none of these were tested in any of the analyses. It would be interesting and relevant to test such factors.
- Measures section: Please describe the measures in more detail (e.g., what does the scale/sub-scale measure, provide a sample item, how did the scale score in internal consistency/cronbach's alpha)
- Line 172: It is stated that low scores represent higher degrees of attachment anxiety. Was this reverse-scored in the analyses? The results suggest that positive correlations equate to higher degrees of attachment anxiety and other test variables. However, if low scores represent higher attachment anxiety, the opposite would be true. Could you please clarify.
- Statistical Analyses section: Were there any missing data and how was it addressed? Same with outliers.
- Line 235-240: Did the results show partial or full mediation in the model? From what I can see, I assume analyses only revealed partial mediation. Please specify more clearly.
- Discussion section: This section is lacking in discussion of the implications of the results and a 'so what?' factor. It would be beneficial to flesh out the discussion section further.
The authors do well to investigate an under-examined issue in adolescent sexual and mental wellbeing. However, further amendments are required to boost the novelty and quality of the paper.
Reviewer 2 Report
Thank you for the opportunity to review this manuscript. Overall, the manuscript addresses interesting questions and is well-written. In order to maximize the contribution of the manuscript, however, it would benefit from the following revisions:
- The authors do not sufficiently identify the gaps in the research literature or establish the contribution of their study (and the selected sample) to the broader literature.
- The words “2. Materials and Methods” are errors in The Figure 2.
- Whether there are the reciprocal effects among three variables ASQ, BSI and MSQ? The current study conducted cross-sectional designs and did not exam the reciprocal effects among variables.
- The authors could plot predicted MSQ against BSI, separately for male and female to represent the interaction between BSI and gender.
Round 2
Reviewer 1 Report
Thank you for the amendments and clarification to questions asked. No further suggestions/feedback.